# The Importance of Gender-Related Anticancer Research on Mitochondrial Regulator Sodium Dichloroacetate in Preclinical Studies In Vivo

**DOI:** 10.3390/cancers11081210

**Published:** 2019-08-20

**Authors:** Donatas Stakišaitis, Milda Juknevičienė, Eligija Damanskienė, Angelija Valančiūtė, Ingrida Balnytė, Marta Maria Alonso

**Affiliations:** 1Laboratory of Molecular Oncology, National Cancer Institute, 08660 Vilnius, Lithuania; 2Department of Histology and Embryology, Medical Academy, Lithuanian University of Health Sciences, 44307 Kaunas, Lithuania; 3Department of Pediatrics, Clínica Universidad de Navarra, University of Navarra, 55 Pamplona, Spain

**Keywords:** sodium dichloroacetate, cancer, preclinical research, gender differences

## Abstract

Sodium dichloroacetate (DCA) is an investigational medicinal product which has a potential anticancer preparation as a metabolic regulator in cancer cells’ mitochondria. Inhibition of pyruvate dehydrogenase kinases by DCA keeps the pyruvate dehydrogenase complex in the active form, resulting in decreased lactic acid in the tumor microenvironment. This literature review displays the preclinical research data on DCA’s effects on the cell pyruvate dehydrogenase deficiency, pyruvate mitochondrial oxidative phosphorylation, reactive oxygen species generation, and the Na^+^–K^+^–2Cl^−^ cotransporter expression regulation in relation to gender. It presents DCA pharmacokinetics and the hepatocarcinogenic effect, and the safety data covers the DCA monotherapy efficacy for various human cancer xenografts in vivo in male and female animals. Preclinical cancer researchers report the synergistic effects of DCA combined with different drugs on cancer by reversing resistance to chemotherapy and promoting cell apoptosis. Researchers note that female and male animals differ in the mechanisms of cancerogenesis but often ignore studying DCA’s effects in relation to gender. Preclinical gender-related differences in DCA pharmacology, pharmacological mechanisms, and the elucidation of treatment efficacy in gonad hormone dependency could be relevant for individualized therapy approaches so that gender-related differences in treatment response and safety can be proposed.

## 1. Introduction

Sodium dichloroacetate (DCA) is an investigational medicinal product which has been employed in experimental and clinical research. It has a potential anticancer preparation, derived from in vitro and animal research, as a metabolic regulator in cancer cells, and it has been proposed to cause multiple changes in cancer cells such as reversing the glycolytic cell phenotype [1,2,3,4].

The main known DCA pharmacological mechanisms are as follows: (1) the principal site of DCA action is the mitochondrial pyruvate dehydrogenase complex (PDC), where inhibition of pyruvate dehydrogenase kinases (PDK) keeps PDC in the unphosphorylated active form, facilitating the oxidative removal of pyruvate and the mitochondrial oxidation of glucose. Sodium dichloroacetate may help stabilize the PDC escaping pyruvate dehydrogenase deficiency and DCA treatment results in decreased lactic acid generation in the tumor microenvironment [1,2,5,6]. (2) Sodium dichloroacetate treatment is accompanied by an increase in reactive oxygen species (ROS) generation leading to a number of pro-apoptotic changes in cancer cells [7]. (3) Sodium dichloroacetate works as Na^+^–K^+^–2Cl^−^ cotransporter (NKCC) inhibitor, and such a pharmacological effect could be important in anticancer efficacy [8]. (4) There are preclinical research reports on the synergistic effects of DCA combined with different drugs or radiation therapy on cancer by reversing resistance to chemotherapy or radiation and promoting cancer cell apoptosis, or by increasing or reducing the cytotoxicity of anticancer agents [9,10,11].

Sodium dichloroacetate has been assigned for clinical research as it received the designation as an orphan medicine for the continuous therapy of rare diseases such as congenital lactic acidosis, malaria, homozygous familial hypercholesterolemia, conditions related to acquired lactic acidosis, and pulmonary arterial hypertension as a hyperproliferative disease [12].

The results of a few small clinical trials suggest that the chronic administration of DCA is usually well tolerated and might have effects against some human cancers. Doses up to 6.25 mg/kg of body weight twice daily can be administered safely, but no controlled clinical trials of DCA in cancer patients are available. Thus, the efficacy and safety of advanced or refractory solid tumors treated with DCA can be neither supported nor negated and requires further study [13,14,15].

The regulatory guidelines for pharmaceutical studies require the evaluation of gender differences, indicating that the medicinal product development process should provide adequate information about the efficacy of medicines in both genders [16,17].

This literature review focused on the growing number of DCA preclinical studies in the cancer area in vivo. Preclinical gender-related and gonad hormone-dependent DCA treatment efficacy elucidation could be important for an individualized therapy approach so that gender-related differences in treatment response and safety can be predicted.

## 2. Methods

A keyword search was conducted in the PubMed database: “dichloroacetate”, “cancer”, “preclinical”, “male”, “female”, “PDK”, “PDC”, “ROS”, “NKCC1”, “pharmacokinetics”, “hepatocancerogenesis”, “side effects”, and a combination of keywords of the study points. The search included studies in vivo of any animal species and their experimental models. Sodium dichloroacetate experimental monotherapy and its combination efficiency with authorized therapy concerning human xenograft cancers in male and female animals were retrieved. The data on sex-related mitochondrial function dimorphism, mitochondria-related genes expression, and DCA metabolism-related factor differences in intact males and females were examined as factors that could have an impact on gender-related DCA pharmacology and anticancer mechanisms. The article type searched for was peer-reviewed publications and the language was English. The reference search was repeated on newly identified items until no additional relevant article was found.

## 3. Gender-Related DCA Effect on Pyruvate Dehydrogenase Deficiency

Sodium dichloroacetate is a structural analog of pyruvate, which, by inhibiting PDK, stimulates pyruvate dehydrogenase (PDH) activity. It inhibits all isoforms of PDK keeping the mitochondrial PDC (made up of E1α, E1β, E2, and E3 PDH subunits) in the active form [18,19,20]. The E1α subunit (PDHA1) gene is located on the X chromosome [21]. Such a gene location has different consequences for males and females with PDHA1 deficiency and may cause gender-related clinical problems mainly depending on the amount of residual PDH enzyme activity [22]. The PDC activity is tightly regulated through phosphorylation of PDH1 by deactivated PDK and activated phosphatase, and changes in PDC enzymatic activity which deactivates PDH1 renders the entire complex inactive [23]. The PDH is present in all tissues, and its deficiency is a cause of primary or acquired lactic acidosis which depends on the equilibration of pyruvate with lactate [24,25]. Sodium dichloroacetate treatment may help stabilize PDC escaping PDH deficiency, which leads to differences in mitochondrial activity from chronic DCA exposure. Sodium dichloroacetate reactivates PDC and consequently promotes oxidative metabolism in the mitochondria: the metabolism of pyruvate shifts from glycolysis towards oxidative phosphorylation, lowering lactate production, and this has been exploited in lactic acidosis treatment [4,25,26].

The PDK N-terminal domain contains a binding site for pyruvate and for its analog DCA; DCA, pyruvate, nicotinamide adenine dinucleotide (NAD^+^), and acetyl-CoA (CoA), a two-carbon acetyl unit, exert substrate activation of PDC by inhibiting PDK [1]. Repeated DCA dosing leads to a sustained increase in the PDC activity [18,27].

The PDC activity was higher in young females than in corresponding male mice and decreased after ovariectomy but not after orchidectomy, indicating that PDC activity depends on female steroids; gender differences in brain mitochondrial respiration are suppressed with a decline in the level of ovarian steroids in mice, and this supports the suggestion that female steroids have an impact on glycolysis [28]. Others have reported that female gonad hormone deficiency decreases PDC activity in mice [29]. Administration of 17β-estradiol to ovariectomized female rats increases the PDC subunits expression and modifies PDC phosphorylation processes [30,31,32]. Adult male rats showed a higher expression of PDC-associated genes compared to females [33].

## 4. Oxidative Phosphorylation of Pyruvate and Gender

Mitochondria play a major role in bioenergetics in mammalian cells. The PDH active state is linked to the mitochondrial energy generation process through the tricarboxylic acid (TCA) cycle generating acetyl-CoA and adenosine triphosphate (ATP). Female gonadal steroids can influence the activity of some TCA cycle enzymes [34]. Compared with young males, higher pyruvate oxidation rates in young female mice were observed, and a decline in ovarian steroid levels was concomitant with decreased PDC activity, but this was not so in males after orchidectomy [28,29]. When cells are affected by PDH deficiency, the amount of pyruvate accumulated and adenosine diphosphate (ADP) generated is reduced: cells are dependent on carbohydrate aerobic oxidation. The PDK inhibition by DCA consequently diverts more pyruvate into the TCA cycle [1,35]. Oxidative phosphorylation (of pyruvate by the mitochondrial respiratory chain, which is composed of five complexes—complex I through complex V (C I–C V)) is the major pathway of ATP synthesis [36]. The PDC catalyzes the step generation of cofactors nicotinamide adenine dinucleotide (NADH) and flavin adenine dinucleotide (FADH2): NADH-linked respiration depends on the activities of PDC, TCA cycle enzymes, C I, C III, and C IV; FADH_2_-linked respiration depends on the activities of C II, C III, and C IV and the efficiency of the phosphorylating apparatus generating ATP from ADP and inorganic phosphate [28,37,38,39].

Gonadal hormones modulate mitochondrial respiratory chain biogenesis [40,41]. Estrogen in neurons enhances mitochondrial bioenergetics and sustains aerobic glycolysis and TCA cycle-driven ATP production [42]. There are sex differences of mitochondrial function in brain cells: the NADH-linked respiration rate was higher in young female than in matched male mice; it was higher in females when the respiratory donor substrate pyruvate was used [28]. Others reported a decrease of C I+III, C II, and C IV activities in mitochondria isolated from female mice cortices 28 days after ovariectomy [43] and a decrease of the C IV activity in mice mitochondria isolated from the brain 90 days after ovariectomy [29]. A higher expression of proteins associated with C III–C V of oxidative phosphorylation and the rate of mitochondrial oxygen consumption in the heart of old female monkeys as compared to age-matched males was reported. [44]. Others noted that only complexes I and IV were significantly different between the genders: they were higher in female as compared to male rats [33]. Sodium dichloroacetate treatment increased the NAD^+^/NADH ratio, and this is consistent with the enhanced C I activity leading to the increased oxidation of NADH generated during glycolysis and TCA cycle. There is an inverse relationship between lactate/pyruvate and NAD^+^/NADH ratios [45,46]. Male murine aortas show higher activity of C IV (cytochrome c oxidase) than in females, and orchiectomy abolishes the gender difference, whereas testosterone administration induces an accretion of C IV RNA, protein, and its elevated activity [47]. The higher expression of genes associated with oxidative phosphorylation in the heart of old female rats as compared to males may suggest a greater oxidative potential of mitochondria in females [33]. Furthermore, mitochondria are the site of the initial step of steroidogenesis: the cholesterol side-chain cleavage enzyme cytochrome P450 converting cholesterol to pregnenolone and the 3β-hydroxysteroid dehydrogenase producing progesterone from pregnenolone are located in mitochondria [48]. Treatment with 17β-estradiol or progesterone of ovariectomized rats enhanced the brain mitochondrial respiration, whereas the coadministration of both did not [49].

## 5. The DCA Effect on the Mitochondrial ROS Generation and Gender

Under physiological conditions, a prime source of ROS is mitochondria: C I and C III are believed to be the major sites for ROS production [50]. The ratio between electrons entering the respiratory chain via FADH2 or NADH determines ROS formation; fatty acid oxidation increases the ratio, whereas it appears to be low during glucose oxidation [51].

The PDK-targeted inhibition increases the production of mitochondrial ROS [5]. Sodium dichloroacetate increases ROS generation in mitochondria [52,53]. Sodium dichloroacetate produces the concentration- and time-dependent increase in the superoxide anion and nitric oxide production in zebrafish mitochondria [54]. It is well known that ROS induce cell cycle arrest and apoptosis; DCA affects cell growth and viability through the ROS production increase derived from the promotion of oxidative metabolism [55,56]. The effects of DCA on multiple myeloma cell viability, cell cycle arrest, and apoptotic cell death were associated with PDK inhibition, restored PDH activity, and the promotion of oxidative metabolism in association with increased intracellular ROS production which depends on the DCA dose [57]. The DCA effect cooperated with C I inhibition promoting the oxidative stress in rat VM-M3 glioblastoma cells [7]. Increased ROS levels in DCA-treated cancer cells were related to the induction of apoptosis associated with the increased cytochrome c expression [58].

Gender differences in hydrogen peroxide production in cardiocyte mitochondria in adult rats were reported: female rats showed lower hydrogen peroxide production in cardiac mitochondria as compared to males [33,59]. The depletion of female hormones by ovariectomy potentially enhanced oxidative damage, whereas orchidectomy did not modify the oxidative stress parameters in mice [28]. The treatment with androgens failed to reverse the orchidectomy-induced oxidative stress in rats, whereas estrogen administration in ovariectomized females reversed it: these data imply the predominant role of female hormones [28,60].

As compared with young male mice, young females have a higher mitochondrial glutathione level, the major antioxidant in the brain [61]. Depletion of female steroids by ovariectomy enhances the oxidative damage concomitantly with the decreased mitochondrial glutathione, but orchidectomy did not modify these oxidative stress parameters in males [28].

Reduction in the activity of the respiratory complexes I–V with age in all rat organs together with the age-dependent increase in oxidative stress and decline in NAD^+^ and ATP content was found; NAD^+^ is an essential electron transporter in mitochondrial respiration [62]. Males have a higher baseline NAD^+^ level than female mice, and ovariectomized females showed a more “male-like” pattern of the NAD^+^ level [63]. An increased NADH/NAD^+^ ratio favors the ROS generation [64]. The reduced electron flow through the inhibition of C I or C III favors the enhanced production of superoxide and H_2_O_2_, and mitochondrial generation of H_2_O_2_ due to the C I inhibition being more relevant than ROS production due to the inhibition of complexes III and IV in situ [65,66,67]. The existence of progressive disequilibrium between ROS generation and the antioxidant protective mechanisms during the aging of both gender rats may be applicable to the human population [62,68].

## 6. The DCA Effect on Na–K–2Cl Cotransporter and Gender

The gender-related DCA effect on differences in the NKCC1 gene (*Slc12a2*) expression in rat thymus was reported: the NKCC1 RNA expression levels in DCA-treated gonad-intact and castrated males were significantly decreased, and no such effect was determined in the gonad-intact and castrated female DCA-treated rats [8,69]. Sodium dichloroacetate treatment caused thymus weight decrease in gonad-intact male and female rats, but no such treatment impact was determined in castrated DCA-treated males and females [69]. This could indicate a synergistic effect of DCA with gonad hormones on thymocyte proliferation [70]. The NKCC1 is a known marker of cell proliferation and apoptosis [71,72,73]; it is distributed in the cells of various tissues, among them, in the thymus [74,75]. Rat thymocyte NKCC1 functional activity is related to the intracellular chloride level which is sensitive to the NKCC1 inhibitor furosemide [76,77,78]. The total NKCC1 protein levels of hypothalamus cells were higher in the embryonic male than in female, and NKCC1 mRNA level was found to be higher in males than in females on the day of birth [79].

The DCA effect on NKCC1 might be related to ROS, as the NKCC activity depends on the oxidative stress [80,81]. Increased formation of ROS is related to the rat thymocyte proliferation inhibition [82]. The DCA-treatment-related reduced thymus weight is linked with the DCA-induced increase of thymocyte percentage in the G_2_–M cycle phase in male rat thymus [70]. The G_2_–M phase arrest in multiple myeloma cell lines induced by DCA was reported, and the described effect was supposed to be related to oxidative stress [57,83]. Sodium dichloroacetate causes ROS-dependent T-cell differentiation [84].

Sodium dichloroacetate treatment caused a significant Hassall’s corpuscle (HC) number increase in the thymus of gonad-intact male rats, and no such effect was found in the gonad-intact females. However, no gender-related difference was found in the castrated rats of both genders [69]. Hassall’s corpuscles represent an indirect differentiation feature of thymus epithelial cells [85,86] and thymocytes apoptosis [87,88]. Thymocytes and thymic epithelial cells express androgen receptors [89].

It was reported that a single DCA dose caused a significantly higher 24 h diuresis in Wistar male rats, and the increased diuresis was related to NKCC2 inhibition [8]. Gender differences in renal NKCC2-related sodium handling have been noted in rats [90,91]. The NKCC2 is more abundant in kidneys of intact females compared to intact males, with a greater transporter density in Sprague–Dawley female rats; ovariectomy suppresses this gender difference and17-β estradiol increases while progesterone decreases NKCC2 abundance in ovariectomized rats; these data support the suggestion that the lower the NKCC2 expression in male rat kidney, the more they are androgen-dependent [92]. The DCA effect on NKCC2 might be related to DCA-induced ROS generation [80,81].

## 7. Preclinical Research of DCA Pharmacology

### 7.1. Pharmacokinetics and Bioavailability

Sodium dichloroacetate is rapidly absorbed from the gastrointestinal tract [93,94]; the absorption is region-dependent: it is rapid in the stomach, duodenum, jejunum, and it is less in the ileum than in the colorectal region [94]. After oral radio-labeled DCA administration to rats and mice, only about 1–2% of the product was found in the feces [95,96]. Sodium dichloroacetate has a specific pharmacokinetics characteristic: double peaks in DCA concentration–time profiles were observed after oral dosing in rats [94,97,98]. The reason for the two absorption peaks was related to gastrointestinal-dependent absorption; the second peak was absent when DCA was administered intravenously [94].

The DCA’s main distribution sites are the liver, muscles, skin, intestines, blood, and kidney; less DCA was found in the lung, testis, spleen, heart, brain, bladder, and fat tissues [93,99], and after 48 h, the tissues contained only 1–2% of the DCA dose [95].

Following transport across the cell membrane via the monocarboxylate transporter and across the mitochondrial membrane via the pyruvate transporter, DCA enters the mitochondrial matrix [100]. Sodium dichloroacetate acts as a substrate and inhibitor of hepatic glutathione S-transferase-zeta1 (GSTζ) [101,102]. Sodium dichloroacetate inhibits its own metabolism: it can irreversibly inactivate GSTζ making itself a mechanism-based self-inhibitor [103,104]. In rats and mice, high concentrations of DCA were found to be incorporated into blood plasma proteins within 3 h of dosing [96].

Pharmacokinetic studies in rodents and dogs demonstrated the clear time-dependent non-linear kinetics of DCA, with a high clearance decrease and high accumulation ratio after repeated dosing [93,98,105]. The DCA distribution and plasma clearance in male animals following repeated dosing vary with species and age [93,94,96,100,105,106,107]. Following the intraperitoneal administration of a single 38.5 mg/kg dose in male F344 rats, the hepatic level of GSTζ immunoreactive protein decreased to less than 40% of the control value: subsequently, 7–8 days were required for the return of the GSTζ protein level and DCA metabolism to control values [19]. Studies in rats have demonstrated that changes in GSTζ activity are directly related to the elimination capacity and persistence of DCA. Following the administration of 50 mg/kg DCA, the elimination half-life was age-dependent: it was significantly shorter in male Sprague–Dawley rats aged 3–4 months than in those aged 16 months. The maximum DCA plasma level following two doses was higher than after a single dose, and the elimination half-life increased after the repeated dosing [93].

During the pharmacokinetics study, Fischer-344 male rats aged 8–10 weeks received 0.05–20 mg/kg of DCA in naïve and GSTζ-depleted rats (GSTζ was depleted by exposing rats to 0.2 g/L DCA in drinking water for 7 days). The GSTζ depletion significantly slowed the elimination of DCA. The oral DCA bioavailability in naïve male rats dosed 5, 20 and 100 mg/kg was significantly lower than in GSTζ-depleted ones (10%, 13%, 81% and 31%, 75%, 100%, respectively). The liver extraction of DCA in the GSTζ-depleted rats had linear kinetics, but it decreased with the metabolism saturation at higher doses [94]. Sodium dichloroacetate is unable to fully inhibit GSTζ activity in rats, and the existence of a portion of DCA intrinsic hepatic clearance free from DCA self-inhibition was suggested [96,104].

The maximum (or peak) serum concentration values (C_max_) that a drug achieves in the body after the drug has been administrated for DCA were greater in male rats relative to those in male mice, while the half-life for DCA in the blood was similar in rats and mice [96]. The elimination of i/v DCA doses in the young (10 weeks) and old (60 weeks) B6C3F1 male mice previously treated with DCA in drinking water showed that DCA immediately reduces metabolism in young male mice to a minimal level, which remains stable till the end of treatment. However, old male mice undergoing chronic DCA treatment had the elimination rate comparable to that of controls, and both young and old treated mice had similar DCA intrinsic metabolic clearances (0.84 and 0.76 mL/h/mg, respectively) [107]. Following DCA treatment, the GSTζ activity in the liver cytosol was reduced by over 80% in male mice aged 10 weeks but it was unaffected in old ones. A decrease in DCA elimination in old mice is consistent with an age-related natural decrease in drug elimination [107]. Sodium dichloroacetate elimination is much slower in male dogs than in mice and rats, possibly because of the striking inhibition of hepatic GSTζ, and dogs have the DCA kinetics similar to that in humans [105,108]. Sodium dichloroacetate metabolism is extensive; only less than 3% of the dose is excreted unchanged in young rats [95,96,109,110], whereas older rats given a dose excreted 7.4–37.2% as the parent DCA [93,100,111].

The preclinical pharmacokinetic studies of DCA were conducted only in males [93,94,96,100,105,106,107,109,112,113]; so, there is no possibility to compare gender-related differences; in some studies, there are no data on the gender of the tested animals [114].

### 7.2. DCA Metabolism

Sodium dichloroacetate is eliminated mainly through GSTζ-catalyzed dechlorination to glyoxylic acid, which is further metabolized by mitochondrial or cytosolic enzymes [97]. Sodium dichloroacetate can also be dechlorinated to monochloroacetic acid in the blood [100,115]. The metabolism of DCA in rodents decreases for prolonged administration [19,109,116,117] as DCA expresses a fast effect in inhibiting its own metabolism after the first dosing, and an increased inhibition was noted after the second oral dosing in male rats by inhibiting GSTζ [103]. In the pharmacokinetic model, measuring plasma DCA concentrations in naive and male rats and mice pretreated with 2 g/L DCA in drinking water, the estimated reduction in DCA metabolism among naive and 2 g/L pretreated rodents was 99% in rats and 76% in mice, showing significant species-related differences in DCA degradation [103]. The rate constants for DCA-dependent GSTζ inactivation in mouse, rat, and human liver cytosol were different as rat > mouse > human [100,116].

Radio-labelled [^14^C]DCA is metabolized to monochloroacetic acid, glyoxylic acid, glycolic acid, oxalic acid, glycine, hippuric acid, carbon dioxide (CO_2_), and chloride anion [96,100]. Sodium dichloroacetate is converted into CO_2_ through its major metabolite glyoxylic acid, and CO_2_ was a major DCA metabolite. The CO_2_ excretion depends on the DCA dose or on its prior exposure in male rats [93,96]. In male rats treated with a single DCA dose, exhaled 26% (fed) and 37% (overnight fasted) as [^14^C]O_2_; in rats treated with two labelled DCA doses 24 h apart, 20% (fed) and 44% (overnight fasted) DCA was recovered as CO_2_ [100,111]. Up to 30% of DCA was expired as CO_2_ in male mice [102]; in others, mice exhaled only 2% of the dose as CO_2_ [96]. The glyoxylic acid, oxalic acid, and glycine in male rat urine accounted for 9–44% of the administered dose, and their excretion depended on the DCA dose and rat age [100]. The urinary excretion of unchanged DCA in male rats increases with age, whereas oxalate and DCA end metabolites show the opposite trend [100]. It is notable that preclinical studies of DCA metabolism are conducted only in male rats [93,95,96,100,109,117,118,119,120,121], male mice [96,122] or male dogs [105], while some others do not indicate the gender of tested animals [99].

Glutathione (GSH)-dependent oxygenation of DCA to glyoxylic acid and its elimination proceed mainly through GSTζ-catalyzed dechlorination in cytosol and mitochondria [102,110,123]. Of cytosolic GSTζ, the ^App^*K*_m_ for GSH obtained using female Sprague–Dawley rats is 2.5 fold higher than in males and 3.2 fold higher than in male Fisher 344 rats; the higher ^App^*K*_m_ for GSH with GSTζ indicates that it has weakened access or binding to GSH [123]. Furthermore, the possible gender and gonad hormone DCA metabolism dependency indirectly support data on gender-related differences of glycolate oxidase participating in the metabolic conversion of glycolic acid to oxalate in the rat liver: the glycolate oxidase activity was significantly lower in intact female and gonadectomized male rats as compared with gonad-intact males; testosterone promotes and estrogen decreases glycolate oxidase activity in male rats. Glycolate increased the urinary oxalate excretion dose-dependently in males but not in female rats [124].

### 7.3. Side Effects of DCA

In animals, DCA has been shown to have side effects such as lowering the blood glucose level [99,125,126,127] and the lactate level in blood serum and tumor tissue [126,128,129,130]. It has numerous toxic adverse effects on the testicular tissue [131] and induced birth defects [54,97], while hepatomegaly was consistently observed across species [97,132,133,134]. The high doses of DCA in male rats exhibited peripheral hind leg neuropathy [133,135]. The DCA-related neuropathy is supposed to be caused by the DCA metabolite oxalate [100,136], the generation of which from glycolic acid is gender- [123,124] and age- [100] dependent, or it is related to tissue thiamine depletion [137]. Sodium dichloroacetate treatment activates PDH with a concomitantly increase de novo CoA biosynthesis in ovarian cancer cells, and induction of cytosolic CoA biosynthesis can aggravate DCA toxicity [138]. However, at present, the relevance of DCA’s adverse effects in an animal to its gender-related human toxicological profile has not been investigated.

### 7.4. DCA Hepatocancerogenic Effect

Sodium dichloroacetate is a genotoxic agent [139,140,141,142]. The Moore and Harrington-Brock review concluded that the available genotoxicity data indicate DCA to be a weakly mutagenic agent causing DNA point mutations and chromosomal aberrations [143]. Sodium dichloroacetate is classified as a carcinogen [133,144].

Carcinogenicity studies in B6C3F1 male mice [135,145,146,147,148,149,150,151,152,153] and in F344 male rats [134,135] show an increase in hepatic adenoma and adenocarcinoma cases after chronic treatment with DCA (Table 1). Sodium dichloroacetate specifically stimulates the growth of hepatoadenomas rather than carcinomas, but adenomas’ malignant transformations cannot be rejected. Sodium dichloroacetate induces liver tumors in male rats at lower doses than in mice, and this strengthens the weight of data for DCA tumorigenicity. Sodium dichloroacetate caused a significantly increased incidence of liver tumors in male mice from different DCA dosages in drinking water [145,147,148,149,150,151,153,154]; the increase in tumor prevalence depends on the DCA concentration and exposure duration [151]. The B6C3F1 male mice studies have established that the ingestion of DCA results in an increased incidence of hepatic adenocarcinomas at the dosages from 0.5 to 5 g/L in drinking water [145,147,148,149,150,151,153,154]: the cancerogenic DCA doses in male mice were found at 1 g/L [150,151], at 2.0 g/L [151], at 3.5 g/L [147,150,151,153,154] and at 5.0 g/L [145,149]; the dose of 0.5 g/L in drinking water, when the exposure was 104 weeks, in male mice induced adenocarcinomas in one study [148], while other researchers did not find such a relationship [151,152].

There are only a few DCA carcinogenicity studies in female mice [146,155,156,157]. The histological findings in female mice indicate the DCA hepatocarcinogenicity response in females to be related to higher DCA doses and longer exposure duration as compared with males (Table 1). However, the female mice carcinogenicity data were collected from studies investigating only females in different laboratories [155,156,157], thus, the data cannot be adequately evaluated for gender-related differences. In mature female mice, a significant increase in tumors was found only at the 3.5 g/L DCA dosage in drinking water for 104 weeks [157], with the comparable tumor frequency in male mice treated with the same dosage and duration [147,150,151].

The significant early-life DCA 3.5 g/L dose’s cancerogenic effect in unmatured (postnatal) mice for 10 weeks treatment, followed by drinking water exposure until 98 weeks of age, was observed both in male and female mice (histology was evaluated at 98 weeks): the tumor incidence comprised 85% and 86%, respectively [154]. It was predicted that short early-life postnatal DCA exposure may alter the cancer risk later in life, resulting in a similar carcinogenic effect in mice livers of both genders via epigenetic-mediated pathways affecting the oxidative cell metabolism [152]. Similar values were reported in matured B6C3F1 male mice (Table 1) treated permanently with 3.5 g/L of DCA in drinking water [150,151].

Miller et al. [158] showed DCA’s effect on liver tumor growth in mice, using magnetic resonance imaging where DCA treatment in mice was prescribed until small tumors became apparent (1 mm in diameter). The animals with tumors were randomly studied in groups: (1) in which treatment was maintained and (2) in which treatment was suspended, where tumors were imaged over the next 2–3 weeks. The cessation of treatment abolished the growth of tumors [158]. Stauber et al. demonstrated that the medium with DCA 0.5 mM increases the proliferation of c-Jun positive hepatocytes; c-Jun is a nuclear transcription factor associated with cell transformation [159]. Studies demonstrate the DCA-induced tumors are c-Jun positive [159,160]. In normal c-Jun negative hepatocytes, DCA treatment consistently inhibits hepatocyte replication in vitro [155,161]. Furthermore, hepatocytes from DCA-induced c-Jun-positive liver tumors are resistant to the DCA inhibitory effect on hepatocyte proliferation [156]. Data from animals treated with DCA show that the dose–time-related hepatocyte proliferation depression was accompanied by a suppression of spontaneous apoptosis [162].

The DCA-exposed mice study suggests that liver tumors can originate from eosinophilic, basophilic, dysplastic, and clear cells [145,163]. The DCA-promoted tumor expresses the CYP2E1 and CYP4A1 of cytochrome P450s [164]. Sodium dichloroacetate inhibiting GSTζ produces hepatocyte cytomegaly due to the glycogen deposition related to the increased tyrosine metabolites level, and this supports the hypothesis that the tyrosine metabolites may be cancerogenic agents [133,147,148,151]. We did not find data which compare the gender-related difference of the DCA metabolism in relation to tyrosine metabolites level. The DCA dose–response related to glycogen deposition in hepatocytes is similarly required for inducing hepatocarcinogenesis [165]. Glycogen deposition could not be replicated by exposing mice to the metabolites of DCA glycolate, glyoxylate, or oxalate in drinking water [166]. Furthermore, the GSTζ deficiency is related to an increased susceptibility to the oxidative stress and activation of stress–response pathways [107,167,168]; oxidative stress is also closely associated with tumor promotion mechanisms [169].

## 8. Experimental Research of DCA Monotherapy Efficacy in Cancer In Vivo

Experimental human cancer xenograft studies in vivo with the DCA treatment monotherapy has been found to have antitumor properties in non-small cell lung cancer [83,170,171], lung adenocarcinoma [172], breast adenocarcinoma [171,173], colorectal cancer [174,175], neuroblastoma [176], glioblastoma [177], pancreatic cancer [178,179], hepatocellular carcinoma [53,180], and ovarian adenocarcinoma [181]. Other researchers found that DCA treatment did not exert antitumor activity against xenografted non-small cell lung cancer [170,182], breast adenocarcinoma [9,183], colon and colorectal cancer [184,185,186,187], and glioblastoma [188,189] in vivo (Table 2). Furthermore, others have reported colorectal cancer xenografted tumors to show a significant growth enhancement after treatment with the DCA in vivo [184].

There are in vivo studies indicating no efficacy of DCA treatment on xenografted animal tumors: the DCA treatment did not impact on the apoptosis of rat mammary adenocarcinoma cells [190]. The DCA treatment promotes Neuro-2a cell tumor (murine neuroblastoma) growth in a female mouse model bearing Neuro-2 tumors [191].

The limitations of several studies should be stressed, and in several cases, they could be evaluated as some kind of inappropriate approach to investigation. As shown in Table 2, a remarkable number of papers did not indicate the gender of animals used for research, and this concerns 46.7% of the DCA-treated groups described in the manuscripts. It is understandable when the researchers used female animals for breast or ovary cancer studies [9,173,183], but there was no grounded reason for why they chose to conduct the studies on different genders in other cancer localizations. As shown in Table 2, of all investigated animal groups, the sex was not indicated for 47% of cases. Experiments with no DCA treatment efficacy according to the indicated sex of animals in 13 studied groups with different localizations of cancer were as follows: 15.38% in males, 38.46% in females, and 46.15% where the sex of animals was not indicated. An effective treatment was found in 16 groups: it comprised 25% in males, 31.25% in females, and 43.75% when the gender of the animals was not indicated (Table 2); the described weaknesses do not allow for a comparison of the gender-related efficacy results. Furthermore, different DCA dosages or their administration route were used. From the DCA efficacy studies described in 29 papers, the age of animal groups used was not indicated in 33.3% of cases (Table 2).

There are data reporting that the DCA parenteral administration was not effective in decreasing tumor growth. This was the reason indicated for why researchers chose per os administration for the DCA treatment [170]. As we can see from Table 2, I/P administration was used in nine study groups, and the treatment efficacy was found in 44% of them; the per os administration was used in 21 tested groups, and the treatment efficacy was determined in 57% of them.

The DCA oral dosage of 86 mg/kg/day in Fisher female rats aged 10–13 weeks is estimated to result in DCA plasma concentrations up to 1.0 mM, while an additional 200 mg/kg/day I/P increases plasma levels to 1.5–3.0 mM [94,110,192]. There was no change in the number of rat mammary adenocarcinoma metastases in rats in the low-dose DCA group (with 0.5–1 mM of DCA in blood serum), and the high-dose DCA group (1.5–3.0 mM of DCA) showed a significant reduction in lung metastases [190]. Sodium dichloroacetate serum levels up to 1 mM were maintained in patients chronically treated with an oral DCA administration of 25 mg/kg/day [193]. Sodium dichloroacetate administration by oral gavage of 50 mg/kg twice daily is based on published reports: 100 mg/kg mouse/day corresponds to approximately 13 mg/kg in humans [194], which is consistent with the doses used in clinical settings [188]. Nude mice were administered 1.4 g/L DCA in drinking water (corresponding to a calculated 100 mg/kg/day, used in patients) [195]. Studies have shown that the daily consumption of 0.5 g/L DCA contained in drinking water resulted in increased PDH activity in mouse tissue [196] and a serum level of DCA equivalent to that in DCA-treated humans [107,197].

## 9. DCA as a Gender-Related Modulator in Cancerogenesis

Sodium dichloroacetate induces downstream pharmacological effects on tumor growth in vivo by causing a shift in metabolism of cancer cells’ mitochondria, increasing mitochondrial function [171,178,184,189,190,198] and ROS production [83,171], decreasing cell proliferation [171,173,190], and enhancing apoptosis [83,171,198].

Through inactivation of PDH, PDKs divert the pyruvate away from mitochondrial respiration and favor its conversion to lactate or alanine [199,200]. The PDK1 is a target of the hypoxia-inducible factor (HIF-1), as the HIF-1-mediated expression of PDK1 increases lactic acidosis [201]. An association between HIF-1 expression and gender are reported [202,203]. Females are more prone to lactic acidosis frequency, and sex-linked mitochondrial genetic differences could increase susceptibility to the development of lactic acidosis [204]. Sex hormones regulate the plasma lactate level: estrogen upregulates but testosterone tend to downregulate lactate levels [205]. The accumulated lactate could be used by tumor cells as an energetic fuel and appear as the factor responsible for tumor resistance to treatment [192,206,207]. Sodium dichloroacetate suppresses lactate acidosis in the tumor microenvironment [173,208,209]. Hypoxia is a feature of solid tumors associated with resistance to therapy [210,211]. The hypoxic microenvironment stimulates epigenetic changes in gene expression through the HIF-1 transcription factor in tumor cells [185,212]. Genes encoding HIF-1 were reproducibly lower in DCA-treated cells when compared with respective controls [198].

Genetically, GSTζ-deficient mice show lymphocyte depletion not specific to DCA-treated mice [213,214]. The DCA treatment was suggested to induce a stronger immune response to the tumor as an increase in the number of T lymphocytes in tumor tissue [190], which may be related to lactate level reduction in the tumor; a high lactic acid level has been shown to depress the T cell function [207]. Several studies have reported DCA anti-tumor effects to be related to the inhibition of angiogenesis, followed by the reduced tumor vascularity perfusion [171] and metastasis suppression [190,208].

## 10. Experimental DCA Combination with Authorized Therapy

The DCA combination with other medicines in cancer therapy aims to achieve a synergistic therapeutic effect, reducing drug dosage, drug toxicity, and minimizing or delaying the induction of drug resistance. The DCA–arginase combination exhibited enhanced anti-cancer effects in preclinical breast cancer models [9]. The bevacizumab and DCA combined treatment dramatically blocked human glioma stem cell xenograft growth in female mice compared to either medicine alone, and this indicates that combining DCA with antiangiogenic therapy represents a potential treatment strategy [188]. Together with radiotherapy, DCA sensitized glioblastoma cells to radiotherapy, induced cell cycle arrest at the G_2_–M phase, increased the oxidative stress in tumor cells, and improved the survival of glioblastoma-bearing female mice [177]. The PDK2 overexpression and high glycolysis are linked to cisplatin resistance in head and neck cancers; DCA significantly sensitized resistant head and neck cancer cells with PDK2 overexpression to cisplatin in vitro and in xenograft tumor-bearing male mice [215]. Sodium dichloroacetate helps to overcome the sorafenib resistance of hepatocellular carcinoma in male mice [180]; it increased the capecitabine antitumor effects in a mouse B16 melanoma allograft and in human lung cancer xenografts in male mice [182]. Sodium dichloroacetate, by inducing in paclitaxel-resistant lung adenocarcinoma cells intracellular citrate accumulation, led to overcoming the resistance by glycolysis inhibition and P-glycoprotein inactivation in male mice [172]. Sodium dichloroacetate increased multiple myeloma cell line sensitivity to bortezomib, and the combined treatment improved the survival of myeloma-bearing mice whose sex was not indicated in the paper [57].

## 11. Discussion

Defective oxidative phosphorylation has a relationship with mitochondrial function attenuation, which is linked with resistance to cancer therapy. Mitochondria function restoration is a target for cancer therapy [216]. Sodium dichloroacetate enhancing the mitochondrial function reverts the hyperglycolytic nature of cancer cells, and, in this manner, tumor cells obtain an impaired proliferative capacity and become more sensitive to apoptotic signals [217].

Sex-related mitochondrial function dimorphism is known in different organ cells of animals [28,33]. The information on different cell mitochondrial activity among genders at different ages in animals is scant. The study of mitochondria-related genes in the heart myocytes in young, adult, and old male and female Fischer 344 rats revealed differences among genders: adult males showed a higher expression of genes associated with PDC compared to female rats; however, the majority of genes involved in oxidative phosphorylation had a higher expression in old females than in age-matched males [33]. Clayton notes that female and male cells differ in their response to chemical agents. Therefore, preclinical research should not ignore the effects of investigational medicines in relation to gender [218]. Gender differences in the expression of genes associated with the mitochondrial metabolism may indicate the possible involvement of mitochondria in DCA-treatment susceptibility. As shown above, data from DCA pharmacokinetics studies are available only on male animals, and, in the literature, there are no comparative studies testing gender-related differences of DCA metabolism in animals. Thus, DCA pharmacokinetics should be examined in animal species by evaluating gender-related differences because the DCA pharmacokinetics of males cannot be directly scaled to females.

Sodium dichloroacetate efficacy may be dependent on multiple factors, including the ability to metabolize DCA via GSTζ, or the overexpression of different PDK isoforms. The four PDK isoforms are inhibited by different DCA concentrations [219] and they are expressed differently in organs [219,220]. Studies are necessary to determine the PDK isoform expression’s correlation with DCA sensitivity and targeted tumors most sensitive to DCA in order to prognosticate which tumors and which gender are most appropriate for DCA treatment. Sodium dichloroacetate has been shown to reverse resistance to a number of agents when resistance to chemotherapy is mediated through the enhanced glycolytic metabolism [180,188,215,221].

Sodium dichloroacetate treatment is evaluated as an adjuvant to chemotherapy [53,57,180,188,221,222] to promote cancer cell death [215,223,224]. It has been suggested that DCA as a single agent will be less effective and could be used in combination with the therapies that would benefit from an enhanced oxidative metabolism [13].

One of the DCA treatment mechanisms is DCA-induced ROS production. Mitochondria are the main cellular regulators of oxidative stress in relation to antioxidative mitochondrial systems [37] which are also related to gender [29,62]. Thus, it is possible to hypothesize that ROS generation under DCA exposure could be age- and gender-dependent in animals.

The elucidation of DCA impact on cancer cell NKCC1’s functional activity in relation to gender and gonad hormones requires further studies. The induction of the NKCC1 RNA expression suppression by DCA indicates that this effect may be related with the thymocyte intracellular chloride (Cl^−^) concentration; the reduced intracellular Cl^−^ level related to inhibitedNKCC1 could be an important factor in cell proliferation and apoptosis. Researchers suggest that cytosolic Cl^−^ could be one of the key targets for anticancer therapy [71,225]. The NKCC1 plays an important role in the proliferation, apoptosis, and invasion of cancer cells [71,72,73]; cancer with a high NKCC1 expression shows a potential progression of tumor, and NKCC1 is recognized as a cancer therapeutic target [226]. A recent review shows NaCl ions in a tumor microenvironment are involved in cancer progression [227]. Sodium dichloroacetate inducing urine excretion of NaCl ions and causing changes in the extracellular and intracellular Cl^–^ concentration could have an important antitumorigenic effect [71,225,226].

The DCA gender-related different effect on thymus and on thymocyte NKCC1 could also be related to the immune response. Thymus is a valuable model to investigate the effect of experimental medicines on elucidating the “non-visible immunological” effect of medicines in athymic animal models. Thus, the DCA gender-related effect on thymocytes could elucidate a very important sphere, also.

Some researchers have reservations concerning DCA anticancer activity [2]; their disagreement has focused on inequality in anticancer activity between in vitro and in vivo studies [228] and effectiveness concerning the DCA dosage, including the administration route [170,190]. Environmental conditions’ influence on cancer cells in vitro is very different from that of environmental states in vivo. Misunderstanding in assessing DCA effectiveness may be due to the use of unreasonably high DCA levels on investigating drug effects in vitro. For example, some papers present cancer cells studied at very high sodium dichloroacetate levels (30, 50, 60, 80 or up to 120 mmol/L) in the medium [58,175,229,230,231]. It is important to emphasize that such “therapeutic” dosages in vitro work due to the hyperosmolar and hypernatraemic conditions in the media, and such sodium dichloroacetate effects determine first of all the hyperosmolar conditions, and this could be evaluated not as medicine efficacy, but as a misconduct of studies in vitro, as 150 mM and a higher level of sodium in blood serum mean a hypernatremic condition, and the blood serum sodium of 170 mM is a mortal condition for a human [232].

## 12. Conclusions

Research on investigational medicines needs to increase awareness of gender-related differences in preclinical and clinical trial phases of pharmaceuticals development as recommended by scientific and regulatory guidelines. Sodium dichloroacetate therapy and authorized therapy combinations with DCA might offer additional treatment options and be a potential new therapeutic regimen against some cancers, which could help overcome resistance to chemotherapy. Further evaluation of DCA and radiotherapy as well as DCA and authorized chemotherapy combinations as treatment approaches to personalized medicine in relation to gender remain important.

## Figures and Tables

**Table 1 cancers-11-01210-t001:** Preclinical studies of liver adenoma and carcinoma prevalence in male and female animals after exposure to different sodium dichloroacetate (DCA) doses in drinking water for a different period.

#	Animal		DCA Treatment		*N*	Tumor Frequency in Treated Animals (%)		Reference
	Species, Age	Gender	Dosage (g/L)	Duration (weeks)		Adenoma	Carcinoma	
1.	B6C3F1 mice, 4 weeks	Male	5	61	26	**96 ***	**81**	[145]
2.	B6C3F1 mice, 4.3 weeks	Male	2.0	37	11	18.2	0	[146]
			2.0	52	24	8.3	25	
		Female	2.0	52	10	0	0	
3.	B6C3Fl mice, 4 weeks	Male	3.5	60	30	**100**	**67**	[147]
			5.0	60	30	**80**	**83**	
4.	B6C3Fl mice, 4 weeks	male	0.5	104	24	**42**	**63**	[148]
5.	B6C3F1 mice, 8 weeks	male	5.0	76	110	**93**	**74**	[149]
6.	B6C3F1 mice; 4 weeks	male	1.0	104	13	no data	**70.6**	[150]
			3.5	104	33	no data	**100**	
7.	B6C3Fl mice, 7–8 weeks	female	1.0	51	50	15	0	[155]
			1.0	82	50	25	3.6	
			3.0	52	40	35	5	
			3.0	82	40	**84.2**	26.3	
8.	B6C3Fl mice, 7 weeks	female	3.0	31	10	0	no data	[156]
			1.0	31	10	0	no data	
			3.0	52	20	35	5	
			1.0	52	19	15	0	
9.	B6C3F1 mice, 4 weeks	female	0.5	104	25	no data	4.0	[157]
			3.5	104	25	no data	**92.0**	
10.	B6C3F1 mice, 4–4.3 weeks	male	0.5	52	10	10	0	[151]
			1.0	52	10	10	0	
			2.0	52	10	0	20	
			3.5	52	10	50	**50**	
			0.5	78	10	10	0	
			1.0	78	10	20	20	
			2.0	78	10	**50**	50	
			3.5	78	10	50	**70**	
			0.5	100	10	20	48	
			1.0	100	10	**51.4**	**71**	
			2.0	100	10	**42.9**	**95**	
			3.5	100	10	**45**	**100**	
11.	B6C3F1 mice, 6 weeks	male	0.5	52	20	20	5	[152]
			2.0	52	19	52.6	5.3	
12.	B6C3F1 mice, 4 weeks	male	1.0	84	27	48	30	[154]
			2.0	84	27	41	32	
			3.5	84	26	**58**	**73**	
		female	1.0	84	26	**35**	8	
			2.0	84	28	21	11	
13.	B6C3F1 mice, 4 weeks	male	3.5	93	44	**59**	**93**	[153]
			Treated and followed for up to 93 weeks:					
			3.5	4	28	25	**82**	
			3.5	10	55	33	**49**	
			3.5	26	54	41	**59**	
			3.5	52	54	**56**	**65**	
14.	F344 rats, 4–4.3weeks	male	0.5	104	23	17.2	10.3	[133]
			1.6 (average)	60	27	10.7	21.4	
15.	F344 rats, 4–4.3 weeks	male	2.4	45	7	14	0	[134]
			2.4	60	27	26	4	

#: the number of experiments. Adenoma and carcinoma percentage frequency values shown in bold are significant compared with their control in the reference study.

**Table 2 cancers-11-01210-t002:** Experimental studies of DCA monotherapy efficacy in different human cancers in vivo.

#	Animal		Xenografted Human Tumor Cells	Cell Line	DCA treatment			Reference
	Species, Age (weeks)	Gender			Dosage, or Daily Dose, Administration Route	Duration (days)	Efficacy on Tumor Growth	
Lung cancer								
1.	Rats nude	unknown	non-small cell lung cancer	A549	75 mg/L per os (in drinking water)	35	↓ tumor volume	[83]
	Rats nude	unknown	non-small cell lung cancer	A549	75 mg/L per os	84	↓ tumor volume	
2.	Rats nude	unknown	non-small cell lung cancer	A549-ASC1	250 mg/kg twice daily I/P	21	no effect	[170]
	Rats nude	unknown	non-small cell lung cancer	A549-ASC1	75 mg/L per os	21	↓ tumor volume	
3.	BALB/c-nu mice, 5–6 weeks	male	non-small cell lung cancer	A549	1.4 g/L per os	30–35	no effect	[182]
4.	Rats nude	unknown	non-small cell lung cancer	A549	70 mg/L per os (50 mg/kg/day)	28	↓ tumor volume	[171]
5.	Mice BALB/c (nut/nut), 4–6	male	lung adeno-carcinoma	A549/Taxol	0.75 g/L per os	7	↓ tumor volume	[172]
Breast cancer								
6.	Rats nude	unknown	breast carcinoma	CRL-2335	70 mg/L per os (50 mg/kg/day)	10	↓ tumor volume	[171]
7.	Mice BALB/c, 6–8	female	breast adenocarcinoma	MDA-MB-231	100 mg/kg/day per os	24	↓ tumor growth	[173]
8.	Mice SCID, 6–8	female	breast adenocarcinoma	MDA-MB-231/eGFP	112 mg/kg/day per os	114	no effect	[183]
9.	Mice BALB/c nude, 6–8	female	breast adenocarcinoma	MDA-MB 231	156 mg/kg/day per os (gavage)	21	no effect	[9]
Colon cancer								
10.	Mice immune deficient RAG1	unknown	colorectal cancer	SW480	150 mg/kg/day per os	14	↑ tumor growth	[184]
	Mice immune deficient RAG1	unknown	colorectal cancer	LS174T	150 mg/kg/day per os	9	no effect	
11.	Mice nude, 6	male	colorectal carcinoma	HCT116	100 μL of 1 mM, 1 mg/kg/day intraperitoneally (I/P)	50	↓ tumor volume	[174]
12.	Mice NCr nude	female	colorectal carcinoma	HT29	200 mg/kg/day per os	4	↓ tumor growth	[175]
13.	Mice nude, 6–8	female	colon cancer	RKO	50 mg/kg/day I/P	40	no effect	[185]
14.	Mice nude, 6–8	unknown	colorectal cancer	RKO	50 mg/kg/day I/P	15	no effect	[186]
	Mice nude, 6–8	unknown	colorectal cancer	RKOshHIF	50 mg/kg/day I/P	15	no effect	
15.	Mice BALB/c-nu-nu, 6–8	unknown	colon adenocarcinoma	WiDr	150 mg/kg twice daily I/P	5	no effect	[187]
16.	Mice NOD-SCID, 5–8	unknown	N-myc amplified neuroblastoma	SKNBE2 xenograft	2.5 mg/kg/day per os (gavage)	28 (5 days a week/for 4 weeks)	no effect	[176]
	Mice NOD-SCID, 5–8	unknown	N-myc amplified neuroblastoma	SKNBE2 xenograft	25 mg/kg/day per os (gavage)	28 (5 days a week/for 4 weeks)	↓ tumor volume	
17.	Mice BALB/c- SCID, 6–8	female	Glioblastoma	U87-MG	50 mg/kg twice daily per os (gavage)	14	no effect	[188]
	Mice athymic nude, 4–8	female	Glioblastoma	U118-MG	50 mg/kg twice daily per os (gavage)	100	no effect	
18.	Mice BALB/c-nude, 6–8	male	Glioblastoma	U87	70 mg/L per os	39	no effect	[189]
19.	Mice BALB/c nude, 8	female	glioblastoma	U87	150 mg/kg/day per os	28	↓ tumor cell proli-feration.	[177]
Other tumors								
20.	Mice nude	unknown	pancreatic cancer	Su86.86	50 mg/kg/day I/P	15	↓ tumor volume	[178]
21.	Mice Harlan nude; 6	female	pancreatic cancer	fresh pancreatic tumor specimen cells	50 mg/kg/day I/P	28	↓ tumor volume	[179]
22.	Mice BALB/c nude, 6–8	male	hepatocellular carcinoma	Hep3B	0.5 g/L, 100 mg/kg/day per os	21	↓ tumor volume	[180]
23.	Mice BALB/c nude, 5–6	male	Hepatoma	HCC-LM3	0,75 g/L per os	35	↓ tumor volume	[53]
24.	Mice nude, 6	female	ovarian adenocarcinoma	SKOV3	50 mg/kg/day I/P	8	↓ tumor volume	[181]

#: the number of experiments.

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
