# Peer review of "The Importance of Gender-Related Anticancer Research on Mitochondrial Regulator Sodium Dichloroacetate in Preclinical Studies In Vivo"

_cancers, 2019, doi:10.3390/cancers11081210_

Round 1

Reviewer 1 Report

The paper presents a very large and exhaustive analysis of Sodium dichloroacetate (DCA) and discuss its anticancer role as  metabolic regulator in cancer cells mitochondria.

The specific aim of this paper is to underline the many gender-related differences in DCA  pharmacology/ pharmacological mechanisms already demonstrated in basic and preclinical research and to discuss, on this basis, the possible gender-related effects on treatment response and safety.

Valuable contributions from this paper are:

-the paper points out on the need to integrate sex as a relevant variable in all the drug developmental phase; currently this integration is strongly  recommended by the International Medicinal Agencies but is not mandatory to provide gender related data at the time of a medicinal marketing authorisation.

-the paper is original since few similar exenstive analysis are available in literature up to now and it could be very useful in order to increase Agencies' and researchers' awareness on the potential importance of considering gender-related effects of human drugs  thus ameliorating the therapeutic approach in both sexes.    (https://www.thelancet.com/journals/lancet/article/PIIS0140-6736(17)30343-4/fulltext)

-the paper include update literature analysis up to 2018 and covering  many relevant aspects (the sex-related mitochondrial functions, the in vivo animal effects  such as DCA pharmacokinetics, the DCA pharmacological effects on tumor growth in vivo also in combination with other drugs.

-concepts are clear and convincing. 2 tables have been included. The main message from these tables is that in many cases the gender related effects remain unknown because females are not included in the experimental population.

Comments and suggestions

-the paper doesn't include a methods section. I suggest to add (in a dedicated section or in the general text) more details on the literature search and an anticipated lists of the topics that have been selected for the search. Even if this is not a systematic revue, these details could improve the significance of the paper;

-I suggest to underline both in the discussion and in the conclusion the contribution of this paper to increasing the awareness on the need to consider gender-related differences as a crucial variable in all the phases of pharmaceuticals development as recommended at scientific and regulatory level.

Author Response

Dear Reviewer,

We would like to thank you for the remarks regarding the manuscript. Please find our answers to your notes

Comments and suggestions

-the paper doesn't include a methods section. I suggest to add (in a dedicated section or in the general text) more details on the literature search and an anticipated lists of the topics that have been selected for the search. Even if this is not a systematic revue, these details could improve the significance of the paper;

Answer. The requirement has been fulfilled. The Method section was included according to the Reviewer’s request.

I suggest to underline both in the discussion and in the conclusion the contribution of this paper to increasing the awareness on the need to consider gender-related differences as a crucial variable in all the phases of pharmaceuticals development as recommended at scientific and regulatory level.

Answer. The Reviewer's note has been fulfilled by amending the conclusions.

We hope this amendment follows from the manuscript and the Discussion, so avoiding the overlapping of the text no amendment was made in the Discussion.

We are thankful for the Reviewer’s remarks an hope that the corrected manuscript meets the requirements of the journal.

Reviewer 2 Report

Extensive English language editing is required before this manuscript is publishable.

Content is novel, interesting and relevant.

Author Response

Dear Reviewer,

We would like to thank you for the remarks regarding the manuscript. Please find our answers to your notes.

Comments and Suggestions for Authors

 Extensive English language editing is required before this manuscript is publishable

 Answer. Great thanks for the Reviewer's remarks. The manuscript language was corrected according to the Reviewer's suggestion.

We are thankful for the Reviewer's remarks an hope that the corrected manuscript meets the requirements of the journal.

Reviewer 3 Report

There is no question that gender affects susceptibility to drug treatment like anticancer drugs, although this has not systematically been analyzed. Hence, a review or perspective that summarizes what we know would indeed be timely. However, there are many other traits that also, and possibly more dramatically, affect susceptibility, such obesity. The present version of the review views everything through the lens of DCA and gender without considering alternative arguments or put the current literature in a broader context. Overall, I am not convinced that the present version offers a sufficiently clear and rigorous treatment of the subject.

1. Perhaps most problematic for this reviewer is the clarity and organization of this review. It is not clear from the beginning what are the premises to expect a different response to DCA treatment on the basis of gender. This becomes somewhat more clear only during the individual sections, but arguments supporting a gender-related effect of DCA are difficult to find. The authors often hop back and forth to the same arguments, and as a results the logic is hard to follow.

2. The review would benefit from a more clear and streamlined presentation of the arguments and evidence supporting them. For example it is nether really clear why DCA in particular should exhibit gender-related effect. What makes DCA mode of action so prone to exhibit differences related to gender? IS DCA different from other treatments? If so what makes so special that one expect gender to have such a big influence on the treatment outcome?

3. The authors should state more clearly what are the expectations/premises of gender related effects, and clarify which of these are supported by experimental evidences and which are not.

4. For example authors describe toxicity-side effects of DCA, but it is no clear how and why this should relate to gender. In fact the authors conclude that so far no study have systematically investigated this aspect. If so it is important to discuss why it can be important to investigate gender-related toxicity, what would be the expectations and possible applications in the clinic. Same argument holds true in many sections of this review. In many sections the relation with gender is totally unclear and often not even mentioned.

5. Recent studies have unveiled a much more complex mechanism of action of DCA (here few examples: PMID:30271981, PMID:25544776, https://doi.org/10.1177/1559325818811522, …) That should be discussed and perhaps could support some of the arguments made by the authors of gender-related effects

Author Response

Dear Reviewer,

We would like to thank you for the remarks regarding the manuscript. Please find our answers to your notes.

Comments and Suggestions for Authors

Moderate English changes required.

Answer. The manuscript language was corrected according to the Reviewer's suggestion.

There is no question that gender affects susceptibility to drug treatment like anticancer drugs, although this has not systematically been analyzed. Hence, a review or perspective that summarizes what we know would indeed be timely. However, there are many other traits that also, and possibly more dramatically, affect susceptibility, such obesity. The present version of the review views everything through the lens of DCA and gender without considering alternative arguments or put the current literature in a broader context. Overall, I am not convinced that the present version offers a sufficiently clear and rigorous treatment of the subject.

 Answer. The authors agree with the note that obesity is an important factor in carcinogenesis. In this respect, experimental data regarding the association between obesity and carcinogenesis in the treatment of DCA are not available in the literature. The purpose of the review was to highlight the importance of gender-related research which could be relevant for the gender-specific treatment.

 1. Perhaps most problematic for this reviewer is the clarity and organization of this review. It is not clear from the beginning what are the premises to expect a different response to DCA treatment on the basis of gender. This becomes somewhat more clear only during the individual sections, but arguments supporting a gender-related effect of DCA are difficult to find. The authors often hop back and forth to the same arguments, and as a results the logic is hard to follow.

 Answer. Thanks for the note. We hope that the correction by adding the Methods section in the manuscript will better illuminate the organization of the manuscript. We also made corrections to the text to reduce its overlapping.

The review would benefit from a more clear and streamlined presentation of the arguments and evidence supporting them. For example it is nether really clear why DCA in particular should exhibit gender-related effect. What makes DCA mode of action so prone to exhibit differences related to gender? IS DCA different from other treatments? If so what makes so special that one expect gender to have such a big influence on the treatment outcome?

Answer. The regulatory and scientific guideline requirements when investigating animals of both genders are common for all investigational medicines. These guidelines indicate the investigation of gender-related differences to be an important variable in preclinical and clinical pharmaceutical development. In the manuscript, we indicate several aspects why the investigation of DCA treatment effectiveness and safety could be important. One example could be: DCA is a structural analog of pyruvate, which by inhibiting PDK stimulates pyruvate dehydrogenase (PDH) activity. DCA inhibiting all isoforms of PDK keeps the mitochondrial PDC (made up of E1ɑ, E1ß, E2, and E3 PDH subunits) in the active form [18–20]. The E1ɑ subunit (PDHA1) gene is located on the X chromosome [21]. Such gene location has different consequences for males and females with PDHA1 deficiency and may cause gender-related clinical problems mainly depending on the amount of residual PDH enzyme activity [22].“

 3. The authors should state more clearly what are the expectations/premises of gender related effects, and clarify which of these are supported by experimental evidences and which are not.

Answer. An example of the gender-related difference of DCA efficacy could be the DCA effect on NKCC1 expression in rat thymocytes. There are clear gender-related differences: the DCA effect also depends on the gonadal hormone (there is no effect on NKCC1 expression in castrated rat thymocytes of both genders). Such facts indicate the importance to determine gender-related effects in the preclinical and clinical pharmaceutical development of medicines to have a possibility to predict the treatment efficacy and safety.

For example authors describe toxicity-side effects of DCA, but it is no clear how and why this should relate to gender. In fact the authors conclude that so far no study have systematically investigated this aspect. If so it is important to discuss why it can be important to investigate gender-related toxicity, what would be the expectations and possible applications in the clinic. Same argument holds true in many sections of this review. In many sections the relation with gender is totally unclear and often not even mentioned.

Answer. Thanks for the note. The manuscript emphasizes that in the literature DCA pharmacokinetics data are available only for male animals. From the manuscript, it is an obvious need to conduct such research in females also, because the gender-related difference in DCA pharmacokinetics and metabolism could cause a difference in the toxic effects at the same dosage. For example, the DCA metabolite oxalate is believed to be important in the rat peripheral hind leg neuropathy pathogenesis. The gender-related oxalate metabolism differences are known. We agree with the Reviewer note that in the manuscript the relationship between toxicity and gender is unclear, but this is reasonable – the respective investigations were not performed. The manuscript indicates the actuality of such research in the future.

Recent studies have unveiled a much more complex mechanism of action of DCA (here few examples: PMID:30271981, PMID:25544776, https://doi.org/10.1177/1559325818811522, …) That should be discussed and perhaps could support some of the arguments made by the authors of gender-related effects.

Answer. Thanks for the remark. Our answers regarding the indicated publications are as follows:

PMID:30271981. We would like to note that the effect of DCA mentioned in the paper was found when high DCA doses were used (up to 25 mM). It is not possible to deny that such dosages could be related to the hypernatremic effect also (sodium level in a medium 145 mM + 25 mM with DCA). Therefore, we decided to amend the manuscript in the section Side effect of DCA with the following sentence: “DCA treatment activates PDH with a concomitantly increase de novo CoA biosynthesis in ovarian cancer cells, and induction of cytosolic CoA biosynthesis can aggravate DCA toxicity“. (Dubuis S, Ortmayr K, Zampieri M. A framework for large-scale metabolome drug profiling links coenzyme A metabolism to the toxicity of anti-cancer drug dichloroacetate. Commun Biol. 2018 Aug 3;1:101. doi: 10.1038/s42003-018-0111-x. eCollection 2018). We hope that the amendment according to this remark is sufficient. Thank you for the suggestion.

- PMID:25544776: The indicated paper is related to DCA clinical investigations of pro-apoptotic mechanisms in vitro. As our review comprises experimental studies in vivo, we did not include it in the manuscript.

- https://doi.org/10.1177/1559325818811522: The indicated paper is related to the study of DCA effect on the NKCC1 and NKCC2 in rats. This article is cited and this DCA mechanism is described in our manuscript.

We are thankful for the Reviewer’s remarks an hope that the corrected manuscript meets the requirements of the journal.

Round 2

Reviewer 2 Report

See attachment.

Author Response

The reviewers’ remark was as follows:

Request. (x) English language and style are fine/minor spell check required.

Answer.  The request was fulfilled. Thanks for your remark. The corrections after the second round were highlighted in green in the manuscript.

Reviewer 3 Report

The authors addressed all my main concerns.

Author Response

(The authors gave the same response as above.)
